# Ellagic Acid Improves Antioxidant Capacity and Intestinal Barrier Function of Heat-Stressed Broilers via Regulating Gut Microbiota

**DOI:** 10.3390/ani12091180

**Published:** 2022-05-04

**Authors:** Tai Yang, Bifan Liu, Yujie Wang, Xiangying Huang, Zhaoming Yan, Qian Jiang, Qinghua Chen

**Affiliations:** 1Hunan Co-Innovation Center of Animal Production Safety, Changsha 410128, China; taiyang@stu.hunau.edu.cn; 2College of Animal Science and Technology, Hunan Agricultural University, Changsha 410128, China; liubifan@stu.hunau.edu.cn (B.L.); wang2441645956@163.com (Y.W.); h2307604419@163.com (X.H.); yanzmmail@163.com (Z.Y.)

**Keywords:** ellagic acid, heat stress, broilers, intestinal microbiota, growth performance, antioxidant system, intestinal barrier function

## Abstract

**Simple Summary:**

Environmental heat stress easily damages feed intake, intestinal health, and growth performance of broilers. Ellagic acid (EA), a natural polyphenol antioxidant in fruits or nuts, is beneficial to animal health. For the first time, we revealed that dietary EA improved antioxidant capacity and the intestinal barrier function of heat-stressed broilers probably via regulating gut microbiota. Therefore, it is proposed that EA could be exploited as a feed additive to alleviate heat stress-induced oxidative damages in broilers.

**Abstract:**

Heat stress (HS) has been revealed to damage the antioxidant system and intestinal barrier function, which greatly threatens poultry production. The present study investigated the effects of dietary ellagic acid (EA) on the antioxidant system, gut barrier function, and gut microbiota of heat-stressed broilers. Arbor Acres 14-day-old broilers numbering 360 were randomly divided into six groups, including one negative control group (NC) and five experimental groups. The broilers in the NC group were supplemented with a basal diet at a normal temperature (23 ± 2 °C). The broilers in the experimental groups were supplemented with basal diets containing EA at different doses (0, 75, 150, 300, and 600 mg/kg) at HS temperature (35 ± 2 °C). The experiment lasted for 4 weeks. Results showed that dietary EA reduced the corticosterone (CORT), LPS, and diamine oxidase (DAO) levels in the serum of heat-stressed broilers. Additionally, dietary EA improved the antioxidant enzyme activity and mRNA levels of Nrf2/HO-1 in the ileum of heat-stressed broilers. The relative abundances of *Streptococcus*, *Ruminococcus_torques*, *Rothia*, *Neisseria*, *Actinomyces*, and *Lautropia* in the cecum were significantly reduced by the EA supplementation in a dose-dependent manner. Notably, the LPS, DAO, and MDA in the serum were revealed to be positively correlated with the relative abundances of *Rothia, Neisseria, Actinomyces,* and *Lautropia*, while the GSH-px, SOD, and CAT levels in the serum were negatively correlated with the relative abundances of *Ruminococcus_torques, Rothia, Neisseria, Actinomyces, Streptococcus,* and *Lautropia*. Taken together, dietary EA improved the antioxidant capacity, intestinal barrier function, and alleviated heat-stressed injuries probably via regulating gut microbiota.

## 1. Introduction

Global meat consumption will reach 51 million tonnes by 2024, of which poultry is estimated to account for half [1]. Global meat production is concentrated in tropical and subtropical regions. However, heat stress is one of the major environmental stressors in tropical and subtropical regions affecting meat production [2,3,4,5]. Poultry is covered with thick feathers and the optimal temperature for growing broilers is 19–22 °C. Therefore, poultry is very susceptible to heat stress compared to other farm animals, and commercial broilers produce more metabolic heat than other local pure genotypes of broilers. In terms of mechanism, heat stress easily damages the antioxidant system, alters intestinal microbiota [4], and disrupts the intestinal function [6] of the poultry [7,8].

Heat stress was reported to induce intestinal microbial disorders and promote pathogenic colonization [9,10,11]. The microbiota in the chicken gastrointestinal tract influences nutrient utilization, immune system development, and physiological function balance [12]. In addition, the intestinal microbiota can maintain intestinal barrier function by forming a protective micro-ecosystem. Once the intestinal microbiota is disturbed, a variety of pathogens can be found in poultry manure-based organic fertilizers, such as *Actinobacillus*, *Bordetella*, *Clostridium*, *Campylobacter*, *Escherichia coli*, *Globicatella*, *Listeria*, *Salmonella*, and *Streptococcus* [13,14,15,16,17].

Natural plants contain a variety of biologically active ingredients and nutrients that promote animal health and improve animal growth performance. The supplementation of plant-derived additives to the diet has been considered an effective strategy to alleviate heat stress-induced damage by scavenging ROS and improving the antioxidant capacity of animals. Polyphenolic compounds have been widely reported to positively affect animal health [18]. Among them, ellagic acid (EA), a polyphenolic compound found in fruits, vegetables, and nuts, plays an important role against oxidative damage, inflammation, and Gram-positive pathogen infection [19,20,21,22,23]. Dietary EA was reported to effectively alleviate paraquat-induced liver oxidative damage [24] and intestinal damage [25] by regulating oxidative stress in post-weaning piglets. In addition, EA has strong antioxidant properties by scavenging ROS and increasing the antioxidant enzymes of the liver in D-galactose-challenged rats [26]. EA also could ameliorate diquat-induced jejunum oxidative stress in C57BL/6 mice by upregulating the mRNA expression of antioxidant enzymes [27]. A previous study showed that EA could enhance glutathione peroxidase (Gpx) activities and total antioxidant (TAC) capacity, which help improve the quality and viability motility of rooster semen under cryopreservation [28]. Additionally, in a study conducted to determine the effects of EA on laying quail exposed to lead toxicity, the results showed that EA improved the performance parameters, enhanced the antioxidant defense system, and decreased the malondialdehyde (MDA) level [29]. However, reports on the application of EA in poultry have been limited [23]. Whether the EA could exert a beneficial role in heat-stressed broilers is also unknown.

Therefore, the present study explored the effects of dietary EA at different levels on growth performance, intestinal barrier function, antioxidant capacity, and gut microbiota in heat-stressed broilers.

## 2. Materials and Methods

All the experimental procedures including broiler raising and sample collection were carried out following the Chinese guidelines for animal welfare, and all animal experiments were approved by the Animal Care and Use Committee of Hunan Agricultural University (2020-075).

### 2.1. Animals and Experimental Design

This experiment was conducted in the Yunyuan Teaching Practice Base of Hunan Agricultural University (Changsha, China). A total of 360 one-day-old Arbor Acres broilers (one-day-old) were purchased from Hunan Shuncheng Industrial Co., Ltd. (Changsha, China). After a 14-day adaptation period (feeding with the basal diet), 360 broilers were randomly divided into six groups with 6 replicates per group and 10 broilers per replicate. One negative control group (NC) and five experimental groups were involved in the study. The broilers in the NC group were supplemented with a basal diet at a normal temperature (23 ± 2 °C). The broilers in the experimental groups were supplemented with basal diets containing EA of different doses (0, 75, 150, 300, and 600 mg/kg) at HS temperature (35 ± 2 °C). During the experiment, after a 14-day adaptation period (feeding with the basal diet), all the broilers were raised for additional 28 days. The basal diet was formulated by the China Agricultural Standard (NY/T33-2004) in terms of the requirements of all nutrients (Appendix A). The EA (purity ≥ 97%) was provided by Shanghai Yuanye Biological Technology Co., Ltd. (Shanghai, China).

All broilers were housed in 2-tier-cage (70 × 70 × 30 cm per cage), and they had ad libitum access to experimental diet and water throughout the experiment. One wire cage with 10 broilers per cage was considered one experimental unit, and the units were uniformly distributed in the house to minimize environmental effects. In our experiment, to ensure ad libitum feed intake, the broilers were fed sufficient diet, twice per day at 07:30 and 16:30, and had ad libitum water access via pressurized nipple drinkers.

Broilers were also provided with a daily lighting schedule of 24-h light and an environment with relatively constant temperature and humidity (temperature 23 ± 2 °C and humidity 60 ± 10% for the NC group; heat stress temperature 35 ± 2 °C and humidity 60 ± 10% for other groups). Routine immunization and disinfection procedures were carried out to ensure the quality of sanitation in the house. The mortality of birds was recorded once death had occurred.

### 2.2. Sample Collection

The schedule for the treatments and sampling is shown in Figure 1. The methods for the sample collection are described below.

#### 2.2.1. Serum Collection

At the age of 28-day-old and 42-day-old, one broiler from each replicate was randomly chosen and weighed, and the blood sample was collected via the wing vein after a 12 h fast. The blood sample was transferred into 10 mL centrifuge tubes and stood for 5 h, followed by centrifugation at 1000× *g* at 4 °C for 10 min and precipitation. The serum was dispensed and transferred into 1.5 mL centrifuge tubes. The serum was stored in a refrigerator at −20 °C until the subsequent analysis.

#### 2.2.2. Intestinal Mucosa Collection

The entire intestine was taken from each euthanized broiler and divided into two parts (jejunum and ileum) according to the anatomical structure. After briefly rinsing with 0.9% saline, the intestinal segments were dissected longitudinally with scissors, unfolded, and placed on a tinfoil-wrapped ice block, and the intestinal mucosa sample was collected with a clean coverslip. Subsequently, the intestinal mucosa sample was marked (treatment group, replicate number, and sampling date), and stored at −80 °C until subsequent analysis.

#### 2.2.3. Cecum Content Collection

The middle of the cecum was cut with sterilized scissors, and cecum contents were collected into sterile centrifuge tubes, marked, stored temporarily in liquid nitrogen, and then transferred to a −80 °C refrigerator for the subsequent tests.

### 2.3. Measurement of Serum Corticosterone and Rectal Temperature

Serum concentrations of corticosterone (CORT) were determined using the ELISA kit (Huaying Biotechnology Co., Ltd., Beijing, China) according to the manufacturer’s instructions. At the age of 28-day-old and 42-day-old, two broilers per replicate were selected for rectal temperature measurement using a veterinary digital thermometer. The thermometer probe was inserted 2–3 cm into the anus, ensuring that the probe was submerged into the cloaca to record the temperature data until the temperature was stable.

### 2.4. Growth Performance Determination

The broilers were weighed at the start and the end of the experiment. Two-weekly feed intake and body weight gain were measured on a per-replicate basis. Final average daily feed intake (ADFI), average daily gain (ADG), and feed conversion ratio (feed intake/body weight gain) were determined.

### 2.5. Measurement of Serum Biochemical Parameters

Serum biochemical parameters including the levels of urea (UA), glucose (Glu), total bilirubin (T-Bil), total protein (TP), total cholesterol (TC), triglycerides (TG), and the activities of aspartate aminotransferase (AST), alanine aminotransferase (ALT), as well as alkaline phosphatase (ALP) were determined using kits (Shenzhen Mairui Biomedical Electronics Co., Ltd., Shenzhen, China) with an automatic biochemical analyzer (BS-420, MINDRA). In addition, serum diamine oxidase (DAO) levels, and bacterial products lipopolysaccharide (LPS) were determined using the ELISA kit (Huaying Biotechnology Co., Ltd., Beijing, China) with a microplate reader (DR-200BS, Wuxi Huawei Delong Instruments Co., Ltd., Shenzhen, China). The analysis conditions in terms of detection method, wavelength, and analysis type are shown in Appendix A (Appendix A).

### 2.6. Measurement of Antioxidant Parameters in Serum and Intestinal Mucosa

In serum and intestine mucosa, the concentration of malondialdehyde (MDA), and the activities of glutathione peroxidase (GSH-Px), catalase (CAT), superoxide dismutase (SOD) were determined using kits (Nanjing Jiancheng Bioengineering Institute, Nanjing, China) with a microplate reader (DR-200BS, Wuxi Huawei Delong Instruments Co., Ltd., Shenzhen, China).

### 2.7. Real-Time Quantitative Polymerase Chain Reaction (RT-PCR)

The method for total RNA extraction from the jejunum and ileum mucosa samples used TRIzol reagent (Invitrogen, Carlsbad, CA, USA) according to the manufacturer’s instructions. The reverse transcription, real-time PCR, and data analysis were as those referred to in a previous study [24]. The purity and quality of the RNA were evaluated by a spectrophotometer (NanoDrop One, Thermo Scientific, Shanghai, China). The target genes used in this study were designed via Primer 5.0 and NCBI Primer-Blast tool according to RefSeq mRNA and synthesized by Sangon Biotech Co., Ltd. (Shanghai, China) (Appendix A). The expression levels of candidate genes in this study were determined using the cycle threshold (Ct) values following the standard curve method. The fold change for each gene was calculated by the 2^−ΔΔCt^ method.

### 2.8. Analysis of Intestinal Microbial Community

The whole-genome DNA of cecum content samples was extracted using the CTAB method according to QIAamp DNA Stool Kit (Tiangen, Beijing, China). DNA concentration and purity were monitored on 1% agarose gels. The 16S rRNA gene was amplified using the specific primer with the barcode (341F: CCTAYGGGRBGCASCAG, 806R: GGACTACNNGGGTATCTAAT). All PCR reactions were carried out in 30 µL reactions with 15 µL of Phusion^®^ High-Fidelity PCR Master Mix (New England Biolabs, Ipswich, MA, USA) and sequencing. PCR products were mixed in an equidensity ratio. Then, the mixture of PCR products was purified with the GeneJET Gel Extraction Kit (Thermo Scientific) and samples with a bright main strip between 400–450 bp were chosen for further experiments. Sequencing libraries were generated using Illumina TruSeq DNA PCR-Free Library Preparation Kit (Illumina, San Diego, CA, USA) following the manufacturer’s recommendations and index codes were added. The library quality was assessed on the Qubit@ 2.0 Fluorometer (Thermo Scientific) and Agilent Bioanalyzer 2100 system. Finally, the library was sequenced on an Illumina NovaSeq platform and 250 bp paired-end reads were generated. The bio-function of intestinal microbiota was predicted via phylogenetic investigation of communities by reconstruction of unobserved states (PICRUSt) analysis. R package ggplot2 was employed to obtain the Spearman’s rank correlation coefficient and cluster stacking heatmap.

### 2.9. Statistical Analysis

Data were analyzed by using IBM SPSS 26.0 statistical software. The differences between NC and HS groups were evaluated using an independent t-test, and the differences among HS, HS + EA75, HS + EA150, HS + EA300, and HS + EA600 were analyzed using one-way ANOVA followed by Duncan’s multiple-range tests. The data were expressed as mean ± SEM. “*” indicates a significant difference between groups where *p* < 0.05, “**” indicates a significant difference between groups where *p* < 0.01. *p* < 0.05 was considered statistically significant.

## 3. Results

### 3.1. Validation of Heat Stress Model of the Broilers

To validate whether the heat stress was successfully induced, we detected the rectal temperature and serum CORT of the 28-day-old and 42-day-old broilers. The results showed that heat stress increased rectal temperature and serum CORT levels (*p* < 0.01) compared with the negative control group (NC group). The dietary supplementation with EA (150 mg/kg) significantly decreased the serum levels of CORT (*p* < 0.05) (Figure 2B).

### 3.2. Effects of Dietary EA on the Growth Performance of Heat-Stressed Broilers

In our experiment, all broilers were healthy and performed well, no mortality occurred in the NC group. The mortality rate in the HS group was 10% (6 out of 60 chickens). The mortality rate of the birds was below 6% in broilers (14 out of 240 birds) in the dietary EA supplementation groups. We further investigated the effects of EA on the growth performance of broilers under heat stress. As shown in Figure 3, heat stress (35 ± 2 °C) exposure without EA supplementation reduced the average daily feed intake of broilers by 31.85–50.17% and damaged the average daily gain by 25.74–44.29%. Notably, dietary EA (75 and 150 mg/kg) increased the final weight of heat-stressed broilers by 8.70–9.91%. Additionally, dietary EA (150 mg/kg) significantly decreased the feed conversion ratio over 14–25 days (*p* < 0.05) (Figure 3E).

### 3.3. Effects of Dietary EA on Serum Biochemical Parameters of Heat-Stressed Broilers

We further tested the serum biochemical parameters of heat-stressed broilers. Results showed that the HS increased the levels of serum AST, ALT, urea, and T-Bil, while the levels of serum ALT and T-Bil were down-regulated by 150 mg/kg dietary EA supplementation (*p* < 0.01) (Figure 4B,C,G,I). In addition, the levels of TG and Glu were decreased by HS exposure, while dietary 150 mg/kg EA supplementation reversed the Glu changes induced by HS exposure (*p* < 0.05) (Figure 4E,H).

### 3.4. Effects of Dietary EA on Intestinal Permeability and mRNA Levels of Tight Junction Proteins of Heat-Stressed Broilers

HS-induced intestinal barrier damage was evidenced by the improved serum DAO level and serum bacterial product lipopolysaccharide (LPS). Serum DAO level of the heat-stressed broilers was markedly decreased by the dietary EA (150, 300, and 600 mg/kg) supplementation (*p* < 0.05), and serum LPS level was significantly decreased by dietary EA (150 and 300 mg/kg) supplementation (*p* < 0.05), compared with the HS group (Figure 5A,B). Meanwhile, the mRNA expressions of ZO-1 and claudin-1 were down-regulated by HS exposure, while dietary 150 mg/kg EA supplementation reversed mRNA level changes of claudin-1 in the jejunum, ZO-1 and claudin-1 in the ileum induced by HS exposure (*p* < 0.05).

### 3.5. Effects of Dietary EA on Antioxidant Enzyme Activity of Serum and Intestinal Mucosa in Heat-Stressed Broilers

Compared with the HS group, dietary EA supplementation (75, 150, and 300 mg/kg) markedly decreased the concentration of MDA (*p* < 0.05) and 150 mg/kg EA increased the activities of SOD, CAT, and GSH-Px in the serum (*p* < 0.05) (Figure 6A). Similar to those serum results, dietary EA (150, 300, and 600 mg/kg) also remarkably decreased the MDA levels, 150 mg/kg EA increased the activity of SOD and CAT in the jejunum (*p* < 0.05), and 150 and 300 mg/kg EA increased the activity of SOD, CAT, and GSH-Px in the ileum (*p* < 0.05) (Figure 6B).

### 3.6. Effects of Dietary EA on the mRNA Levels Involved in the Antioxidant System in Heat-Stressed Broilers

We further investigated the effect of dietary EA on the antioxidant system of the heat-stressed broilers. Results showed that the HS significantly decreased the mRNA levels of ileum Nrf2 and HO-1 (*p* < 0.01), while the adverse effects of HS were alleviated by 150 mg/kg dietary EA supplementation (*p* < 0.05) (Figure 7B,C). At the same time, the mRNA expressions of NQO1, CAT, and MnSOD were up-regulated by dietary 150 mg/kg EA supplementation (*p* < 0.05) (Figure 7).

### 3.7. Effects of Dietary EA on Cecum Microbiota in Heat-Stressed Broilers

#### 3.7.1. Two-Dimensional Nonparametric Multidimensional Scaling (NMDS) Analysis on Cecum Microbiota

Based on Bray–Curtis distance metric data of comparisons among groups, the two-dimensional NMDS (nonparametric multidimensional scaling) ordination plots of cecum microbial communities showed that the cecum microbiota in the HS group exhibited significant difference from those in dietary EA (150, 300, and 600 mg/kg) supplemented groups (Figure 8).

#### 3.7.2. Cecum Microbiota Composition of Heat-Stressed Broilers

Changes in the composition of the intestinal microbiota are closely associated with intestinal health. The top seven abundant bacteria at the phylum level in the cecum are presented in Figure 9A. The top five phyla in the sample were *Firmicutes*, *Bacteroidetes*, *Verrucomicrobiota**, Campilobacterota,* and *unidentified_Bacteria*, followed by *Desulfobacteria* and *Proteobacteria*. Compared with the HS group, dietary EA (150 and 300 mg/kg) markedly reduced the relative abundance of cecum *Actinobacteriota* and *Proteobacteria* (*p* < 0.05). The top seven most abundant bacteria at the family level in the cecum are presented in Figure 9B. Compared with the HS group, dietary EA reduced (*p* < 0.05) the relative abundance of cecum *Streptococcaceae*, *Micrococcaceae*, *Neisseriaceae*, *Lachnospiraceae*, *Burkholderiaceae*, *Actinomycetaceae*, *Pasteurellaceae*, and *Campylobacteraceae*. Then, the top seven most abundant bacteria at the genus level in the cecum are presented in Figure 9C. Compared with the HS group, dietary EA reduced (*p* < 0.05) the relative abundance of cecum *unidentified_Lachnospiracea*, *Ruminococcus_torques*, *Streptococcus*, *Rothia*, *Neisseria*, *Actinomyces*, *Lautropia*, and *Campylobacter* (Figure 9B). Further linear-regression analysis between EA dosage and bacterial abundance verified the dose-dependent regulatory roles of EA on the *Streptococcus*, *Ruminococcus_torques*, *Rothia*, *Neisseria*, *Actinomyces*, and *Lautropia* (Appendix A).

#### 3.7.3. LEfSe Analysis of Cecum Microbiota in Heat-Stressed Broilers

The changes in the profile of the cecum microbiota of the heat-stressed broilers are shown in Appendix A. With LDA value > 4, the LEfSe analysis results showed that order *Lactobacillales* was significantly enriched in heat stress + 300 mg/kg EA group, while order *Lachnospirales* was significantly enriched in the HS group (Appendix A).

#### 3.7.4. PICRUSt Analysis of Intestinal Microbial Functions in the Heat-Stressed Broilers

In this study, we investigated the function alteration of gut microbiota in the heat-stressed broilers with or without EA supplementation using PICRUSt2. The KEGG results showed that replication, recombination, and repair of proteins, glycerophospholipid metabolism, C5-Branched dibasic acid metabolism, metabolism of taurine and hypotaurine were significantly enriched in the EA supplemented group (Appendix A).

#### 3.7.5. Correlation between Intestinal Microbiota (at Genus Level) and Antioxidant Activity or Intestinal Permeability

We performed Spearman’s rank correlation analysis to explore the potential relationship between gut microbiota (at genus level) and the parameters of antioxidant capacity or intestinal permeability of the heat-stressed broilers (Figure 10). On day 28, the abundances of *Lautropia* and *actinomyces* showed a significant positive correlation with the levels of serum MDA, and *Ruminococcus_torques*, *Rothia*, *unidentified_Lachnospiraceae*, *Lautropia*, *Actinomyces*, *Streptococcus*, and *Neisseria* showed a significant negative correlation with the activities of CAT and GSH-Px in serum (Figure 10A). On day 42, the abundances of *Rothia*, *unidentified_Lachnospiraceae*, *Lautropia*, *Actinomyces*, and *Neisseria* showed a positive correlation with the levels of MDA in serum. While the abundances of *Ruminococcus_torques*, *Rothia*, *unidentified_Lachnospiraceae*, *Lautropia*, *Actinomyces*, *Streptococcus*, and *Neisseria* exhibited a significant negative correlation with the activities of SOD and GSH-Px in serum (Figure 10B). On day 28, the abundances of *Rothia*, *unidentified_Lachnospiraceae*, *Lautropia*, *Actinomyces*, *Streptococcus*, and *Neisseria* were positively correlated with the level of LPS. While the abundances of *Actinomyces*, *unidentified_Lachnospiraceae*, *Neisseria*, and *Lautropia* were found to be positively correlated with the levels of DAO (Figure 10B). On day 42, the abundances of *Actinomyces*, *unidentified_Lachnospiraceae*, *Neisseria*, *Lautropia*, and *Rothia* were positively correlated with the levels of DAO, while the *Actinomyces* and *unidentified_Lachnospiraceae* were positively correlated with the levels of LPS (Figure 10B).

## 4. Discussion

Heat stress describes dysfunction of the heat homeostasis including a constant increase in body temperature and respiratory rate, during which the animal’s feed intake, nutrient absorption, and productivity are damaged [11,30,31]. Rectal temperature is a common indicator of heat homeostasis, and it is widely used to measure the response of poultry to environmental stress. In addition, serum CORT level is one of the effective heat stress markers in chickens. In the present study, the high-temperature environment at 35 ± 2 °C significantly increased the rectal temperature and serum CORT levels of broilers, indicating that heat stress had been successfully induced. Interestingly, dietary 150 mg/kg EA supplementation significantly alleviated the increases in CORT level, suggesting that EA might be helpful for heat-stressed broilers.

Heat stress readily induces oxidative damages, which are supposed to be the key mechanisms in the damage to poultry production [5]. Previous studies demonstrated that heat stress can reduce feed intake and body weight gain of broilers by 21.51% and 24.46%, respectively [32]. Consistently, in the present study, we observed that the heat stress environment (35 ± 2 °C) reduced the average daily feed intake of broilers by 31.85–50.17% and decreased the average daily weight gain by 25.74–44.29%. Similar to a recent finding showing that dietary antioxidants improved the growth performance of weaned mice [33], the present study revealed that dietary EA at a dose of 75 and 150 mg/kg increased the final weight gain of the heat-stressed 14-day-old broilers by 8.70–9.91% under HS. Energy is shifted from the production system to the protection strategy against heat stress in heat-stressed birds [34]. We also found that dietary EA at a dose of 150 mg/kg significantly reduced the feed conversion ratio. AST, ALT, and ALP, as biochemical indices that reflect the liver function, can be sensitively detected in the serum [35]. Oxidative stress-induced liver injuries are also reflected by the increases in serum AST and ALT levels [36]. Consistently, the heat stress significantly raised the levels of serum AST and ALT of the broilers, indicating that liver function was disturbed by the heat stress. Interestingly, the increased levels of serum ALT by HS exposure were successfully reversed by the dietary EA supplementation, which indicates that dietary EA is helpful to the liver function of heat-stressed broilers. Following this, we detected the relevant biomarkers of liver function since the liver plays an important role in maintaining the circulation of nutrients and energy metabolism [37,38,39]. The results showed that HS significantly reduced TG, T-Bil, and Glu levels, while dietary EA could effectively elevate the levels of T-Bil, and Glu. Thus, it is implied that dietary EA supplementation could ameliorate the carbohydrate metabolism and alleviate metabolic disorders in heat-stressed broilers.

Heat stress-induced intestinal barrier damage can be evidenced by the increase in intestinal permeability to bacterial products (LPS) [40,41], which were in line with our results that the levels of DAO and LPS in the serum were raised by heat stress. Interestingly, the raised levels of serum LPS and DAO were successfully reduced by the dietary EA supplementation. The disruption of the intestinal tight junction increases intestinal permeability [42]. In the present study, we determined whether dietary EA affects mRNA levels of tight junction proteins. The results indicated that HS exposure decreased the mRNA levels of ZO-1 and claudin-1. As expected, our results showed that EA supplementation significantly up-regulated mRNA levels of ZO-1 in the jejunum and ileum, and claudin-1 in the ileum. These results suggested that the dietary EA may also exert protective roles to the intestinal barrier via regulating mRNA expression of ZO-1 and claudin-1.

Heat stress was reported to disturb the homeostasis of the antioxidant system, thus resulting in elevated ROS levels and reduced antioxidant enzyme activity [43]. It has been revealed that EA reduces the lipid peroxidation marker MDA and activates cellular rescue pathways through the activation of antioxidant enzymes [19]. In this study, the MDA level in broiler serum was significantly increased by heat stress, which indicates oxidative damages were induced by the heat environment exposure. Surprisingly, comparing the heat stress group, dietary EA at different levels markedly decreased the MDA level, indicating a reduction of oxidative damages. Endogenous antioxidant systems in cells are responsible for maintaining ROS homeostasis by scavenging excess free radicals [44]. Given that heat stress may damage the antioxidant system [45], we further detected the levels of the antioxidant enzyme, including SOD, CAT, and GSH-Px. Indeed, our results showed that EA significantly increased the activities of SOD, CAT, and GSH-Px in serum, which is consistent with a previous study [43]. Our data also showed that EA supplementation remarkably decreased the concentration of MDA, and improved the activities of SOD, CAT, and GSH-Px in the jejunum and ileum. These results were consistent with one previous report that the reduction of MDA content in tissues was associated with the enhancement of antioxidant enzyme activity [46]. Taken together, it can be implied that EA could alleviate heat stress-induced oxidative damage via enhancing antioxidant enzyme activity.

Nrf2 is an important transcription factor in the antioxidant system of the body [47], and the binding of Nrf2 to antioxidant elements induces the expression of antioxidant enzymes such as SOD, CAT, and GSH-Px, as well as GSH synthesis [47]. The downstream antioxidant genes of Nrf2, such as HO-1, and NQO1, exert beneficial activity by protecting against oxidative stress [48]. In the present study, results indicated HS failed to damage mRNA expression of Nrf2 and antioxidant enzymes in the jejunum. Similarly, a previous study reported that HS did not affect Nrf2, NQO1 levels in the jejunum [49,50]. Thus, it is surmised that the jejunum is not sensitive to HS exposures. In contrast, the mRNA levels of Nrf2, and HO-1 in broiler ileum were significantly decreased by HS, whereas dietary EA increased Nrf2, HO-1, NQO1, CAT, and MnSOD levels. Previously, the EA metabolite “urolithins” was also reported to upregulate Nrf2 mRNA levels in colon epithelial cells [18]. Therefore, the dietary EA may also enhance the antioxidant system of the intestine via regulating the mRNA expression of Nrf2, HO-1, and the antioxidant enzymes.

Intestinal microbiota has been considered a key target for natural polyphenol regulation of animal health [51]. In this study, EA supplementation indeed modified the microbial composition of the cecum, which prompted us to explore whether the microbiota is involved in the beneficial roles of EA. The disruption of intestinal microbiota homeostasis will result in the colonization of a variety of pathogens in the intestine [12]. The abundance of *Proteobacteria* was reported to be positively related to intestinal disease [52]. In the present study, the phylum level *Proteobacteria* and *Actinobacteriota* were enriched in the heat-stressed broiler, however dietary EA supplementation significantly decreased their abundance. In addition, dietary EA supplementation also decreased the abundances of *Streptococcaceae*, *Micrococcaceae*, *Neisseriaceae*, *Lachnospiraceae*, *Burkholderiaceae*, *Actinomycetaceae*, *Pasteurellaceae*, and *Campylobacteraceae* at the family level. At the same time, dietary EA supplementation also decreased the abundances of *Streptococcus, Ruminococcus_torques*, *Rothia*, *Neisseria, Actinomyces*, and *Lautropia* at the genus level. *Campylobacter* is considered to be closely related to zoonotic campylobacter disease, the increase of [Ruminococcus]_torques_group abundance was associated with the severity of bowel symptoms and seemed to be especially involved in controlling paracellular permeability [53,54]. These results indicated that dietary EA supplementation to heat-stressed broilers may reduce colonization of pathogenic bacteria. The cross-talk between the intestinal microbiota and intestine is critical in preventing detrimental damage induced by heat stress [55]. *Lactobacillus* is a Gram-positive bacterium that contributes to the beneficial microorganism growth on the mucosal surface of animals, which plays an important role in the regulation of oxidative stress [56,57]. LEfSe analysis showed that *Lachnospiraceae* was dominant in heat stress-induced broilers, while *Lactobacillales* was enriched in EA-supplemented broilers (Appendix A). Consistent with these results, microbial function prediction analysis revealed that the functions related to immune system diseases and neurodegenerative diseases were enriched in the gut microbiota of the heat-stressed broilers, while functions associated with lipid metabolism, and carbohydrate metabolism were enriched in EA-supplemented broilers. Interestingly, the relative abundances of microbiota reduced by the EA supplementation were positively correlated with the levels of serum MDA, LPS, and DAO, and negatively correlated with the levels of serum SOD, GSH-Px, and CAT. Therefore, it can be inferred that the modified microbial composition by EA might be involved in the beneficial roles of dietary EA in heat-stressed broilers.

## 5. Conclusions

In conclusion, heat stress negatively affected the final body weight, intestinal barrier function, and antioxidant system of the broilers, while dietary EA exhibited mitigating effects. These beneficial roles of dietary EA might be attributed to profile modification of the gut microbiota in the heat-stressed broilers. In future studies, the mechanism by which interaction between EA and gut microbiota regulates the health of heat-stressed poultry is a suggestion to be investigated.

## Figures and Tables

**Figure 1 animals-12-01180-f001:**
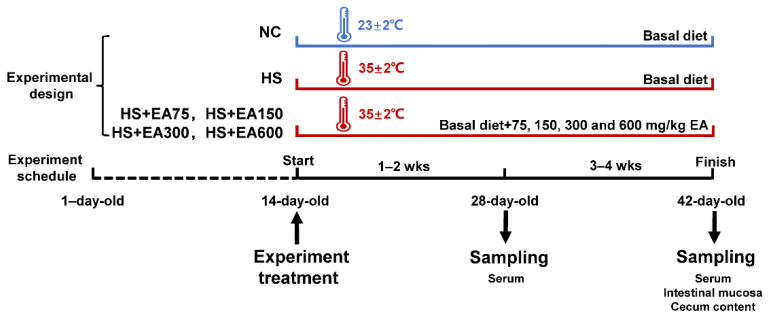
The experimental design and sampling schedule. Abbreviations: NC, negative control; HS, heat stress environment; EA, ellagic acid.

**Figure 2 animals-12-01180-f002:**
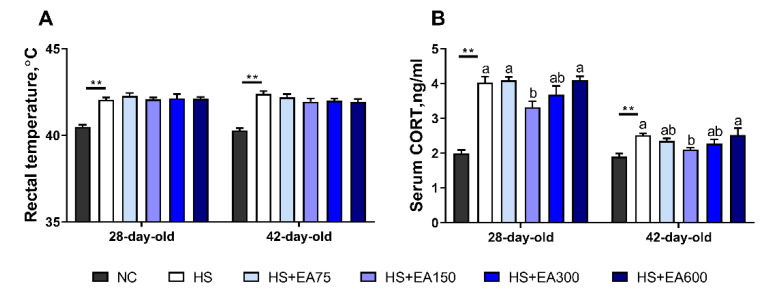
Validation of heat stress (HS) model of the broilers. (**A**) Rectal temperature. (**B**) Serum CORT, corticosterone. Data are presented as mean ± SEM (n = 6). NC (negative control) represents the broilers supplemented with a basal diet at normal temperature (23 ± 2 °C). The broilers in the 5 experimental groups were supplemented with basal diets containing different levels of EA (0, 75, 150, 300, and 600 mg/kg) at HS temperature (35 ± 2 °C), which are represented as HS, HS + EA75, HS + EA150, HS + EA300, and HS + EA600, respectively. Shared superscript letters indicate no significant difference (*p* > 0.05). “**” indicates a significant difference between groups where *p* < 0.01.

**Figure 3 animals-12-01180-f003:**
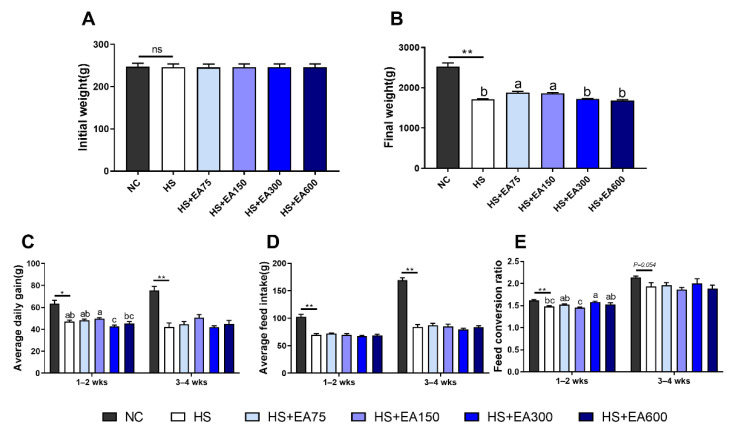
Effects of different levels of EA on the performance of the heat-stressed broilers. (**A**) Initial weight. (**B**) Final weight. (**C**) Average daily gain. (**D**) Average feed intake. (**E**) Feed conversion. Data are presented as mean ± SEM (n = 6). NC (negative control) represents the broilers supplemented with a basal diet at normal temperature (23 ± 2 °C). The broilers in the 5 experimental groups were supplemented with basal diets containing different levels of EA (0, 75, 150, 300, and 600 mg/kg) at HS temperature (35 ± 2 °C), which are represented as HS, HS + EA75, HS + EA150, HS + EA300, and HS + EA600, respectively. Shared superscript letters indicate no significant difference (*p* > 0.05). “*” indicate a significant difference between groups (*p* < 0.05). “**” indicate a significant difference between groups (*p* < 0.01). “ns” indicates a no significant difference between groups where *p* > 0.05.

**Figure 4 animals-12-01180-f004:**
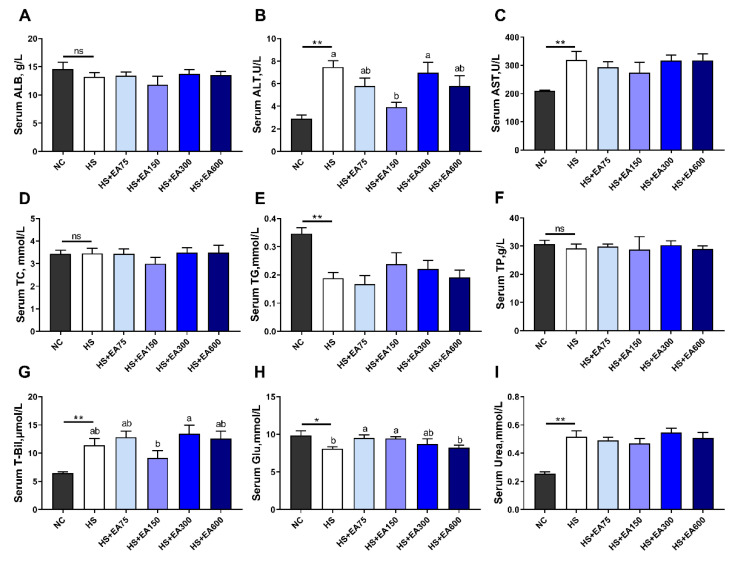
Effects of different levels of dietary EA on serum biochemical parameters of heat-stressed broilers. (**A**) Serum ALB, albumin. (**B**) Serum ALT, glutamic-pyruvic transaminase. (**C**) Serum AST, aspartate transaminase. (**D**) Serum TC, total cholesterol. (**E**) Serum TG, triglyceride. (**F**) Serum TP, total protein. (**G**) Serum T-Bil, total bilirubin. (**H**) Serum Glu, glucose. (**I**) Serum urea. Data are presented as mean ± SEM (n = 6). NC (negative control) represents the broilers supplemented with a basal diet at normal temperature (23 ± 2 °C). The broilers in the 5 experimental groups were supplemented with basal diets containing different levels of EA (0, 75, 150, 300, and 600 mg/kg) at HS temperature (35 ± 2 °C), which are represented as HS, HS + EA75, HS + EA150, HS + EA300, and HS + EA600, respectively. Shared superscript letters indicate no significant difference (*p* > 0.05). “*” indicates a significant difference between groups where *p* < 0.05. “**” indicates a significant difference between groups where *p* < 0.01. “ns” indicates a no significant difference between groups where *p* > 0.05.

**Figure 5 animals-12-01180-f005:**
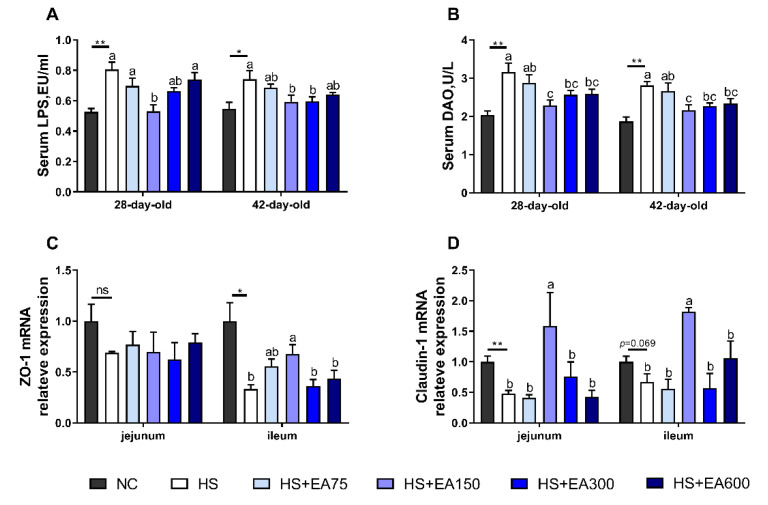
Effects of dietary EA on intestinal permeability markers in serum and mRNA levels of tight junction proteins in the intestine of heat-stressed broilers. (**A**) Serum LPS, lipopolysaccharide. (**B**) Serum DAO, diamine oxidase. (**C**) ZO-1, zonula occludens 1. (**D**) Claudin-1. Data are presented as mean ± SEM (n = 6). NC (negative control) represents the broilers supplemented with a basal diet at normal temperature (23 ± 2 °C). The broilers in the 5 experimental groups were supplemented with basal diets containing different levels of EA (0, 75, 150, 300, and 600 mg/kg) at HS temperature (35 ± 2 °C), which are represented as HS, HS + EA75, HS + EA150, HS + EA300, and HS + EA600, respectively. Shared superscript letters indicate no significant difference (*p* > 0.05). “*” indicates a significant difference between groups where *p* < 0.05. “**” indicates a significant difference between groups where *p* < 0.01. “ns” indicates a no significant difference between groups where *p* > 0.05.

**Figure 6 animals-12-01180-f006:**
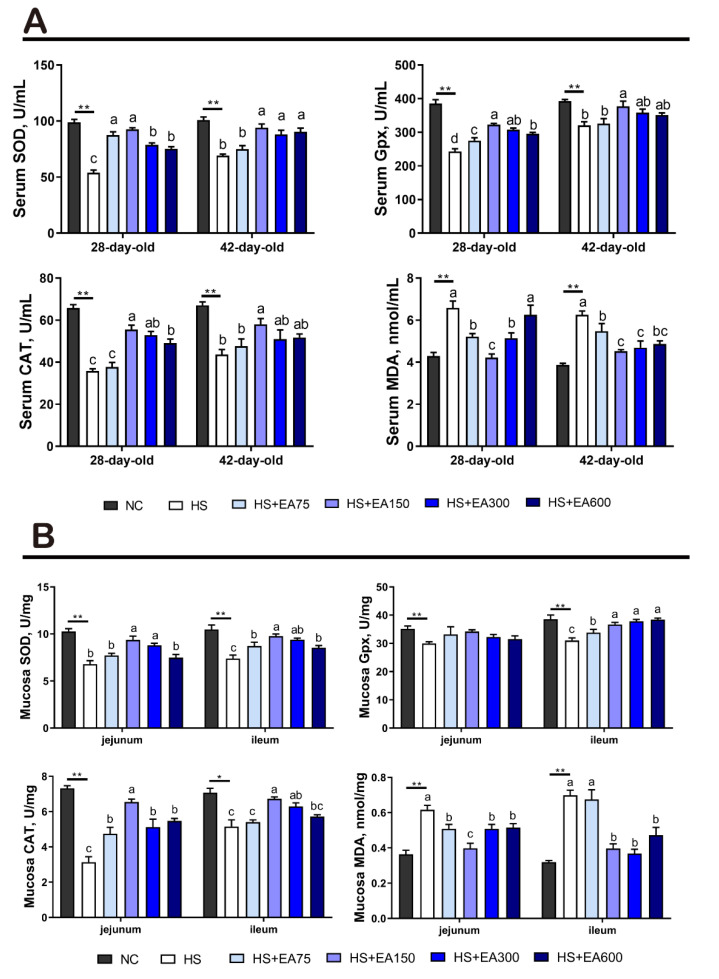
Effects of dietary EA on serum and mucosa antioxidant enzyme activities of the heat-stressed broilers. (**A**) Serum antioxidant enzyme activity; (**B**) antioxidant enzyme activity of the jejunal and ileal mucosa. SOD, superoxide dismutase. GSH-px, glutathione peroxidase. CAT, catalase. MDA, malondialdehyde. Data are presented as mean ± SEM (n = 6). NC (negative control) represents the broilers supplemented with a basal diet at normal temperature (23 ± 2 °C). The broilers in the 5 experimental groups were supplemented with basal diets containing different levels of EA (0, 75, 150, 300, and 600 mg/kg) at HS temperature (35 ± 2 °C), which are represented as HS, HS + EA75, HS + EA150, HS + EA300, and HS + EA600, respectively. Shared superscript letters indicate no significant difference (*p* > 0.05). “*” indicates a significant difference between groups where *p* < 0.05. “**” indicates a significant difference between groups where *p* < 0.01.

**Figure 7 animals-12-01180-f007:**
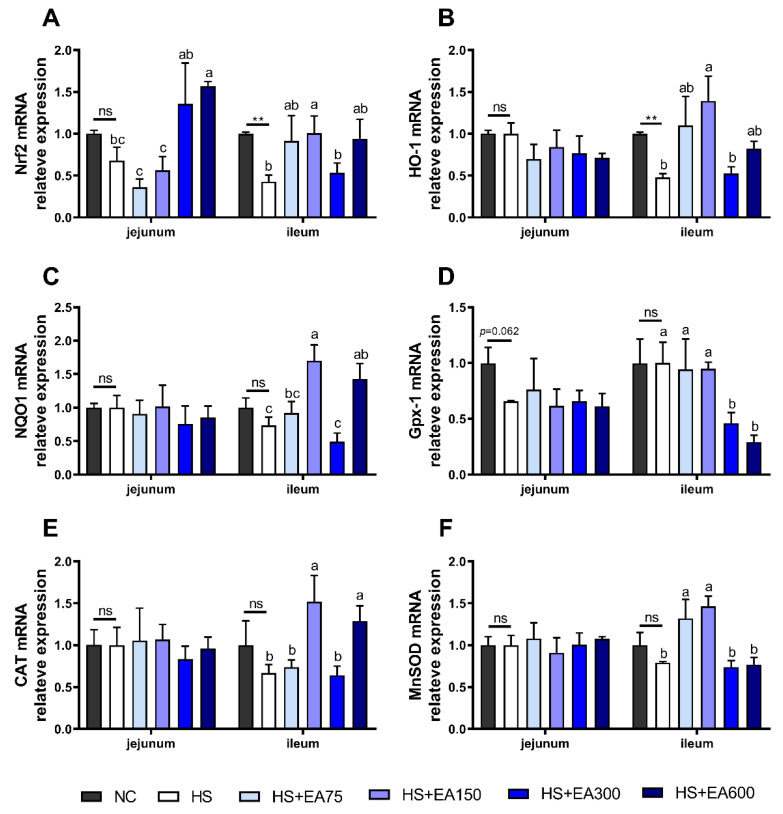
Effects of dietary EA on the mRNA levels involved in the antioxidant system in heat-stressed broilers. (**A**) Nrf2, nuclear factor-E2-related factor 2; (**B**) HO-1, heme oxygenase 1. (**C**) NQO1, NAD(P)H: quinone oxidoreductase. (**D**) GPx, glutathione peroxidase. (**E**) CAT, catalase. (**F**) MnSOD, manganese-containing superoxide dismutase. Data are presented as mean ± SEM (n = 6). NC (negative control) represents the broilers supplemented with a basal diet at normal temperature (23 ± 2 °C). The broilers in the 5 experimental groups were supplemented with basal diets containing different levels of EA (0, 75, 150, 300, and 600 mg/kg) at HS temperature (35 ± 2 °C), which are represented as HS, HS + EA75, HS + EA150, HS + EA300, and HS + EA600, respectively. Shared superscript letters indicate no significant difference where *p* > 0.05. “**” indicates a significant difference between groups where *p* < 0.01. “ns” indicates a no significant difference between groups where *p* > 0.05.

**Figure 8 animals-12-01180-f008:**
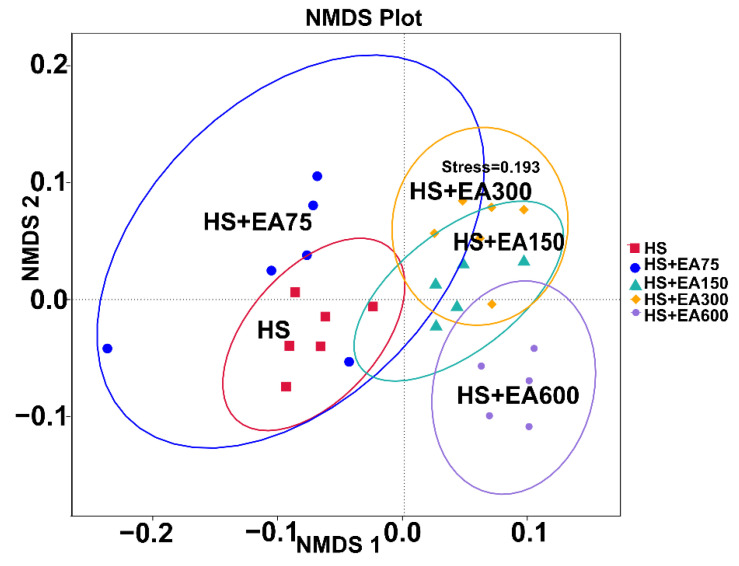
Two-dimensional nonparametric multidimensional scaling (NMDS) ordination plots of cecum microbial communities based on Bray–Curtis distance metric data of comparisons between groups. The broilers in the 5 experimental groups were supplemented with basal diets containing different levels of EA (0, 75, 150, 300, and 600 mg/kg) at HS temperature (35 ± 2 °C), which are represented as HS, HS + EA75, HS + EA150, HS + EA300, and HS + EA600, respectively. Ellipses indicate the standard deviations (SD). Red box represents HS group; blue circle represents HS + EA75 group; green triangle represents HS + EA150 group; yellow diamond represents HS + EA300 group; and purple circle represents HS + EA600 group.

**Figure 9 animals-12-01180-f009:**
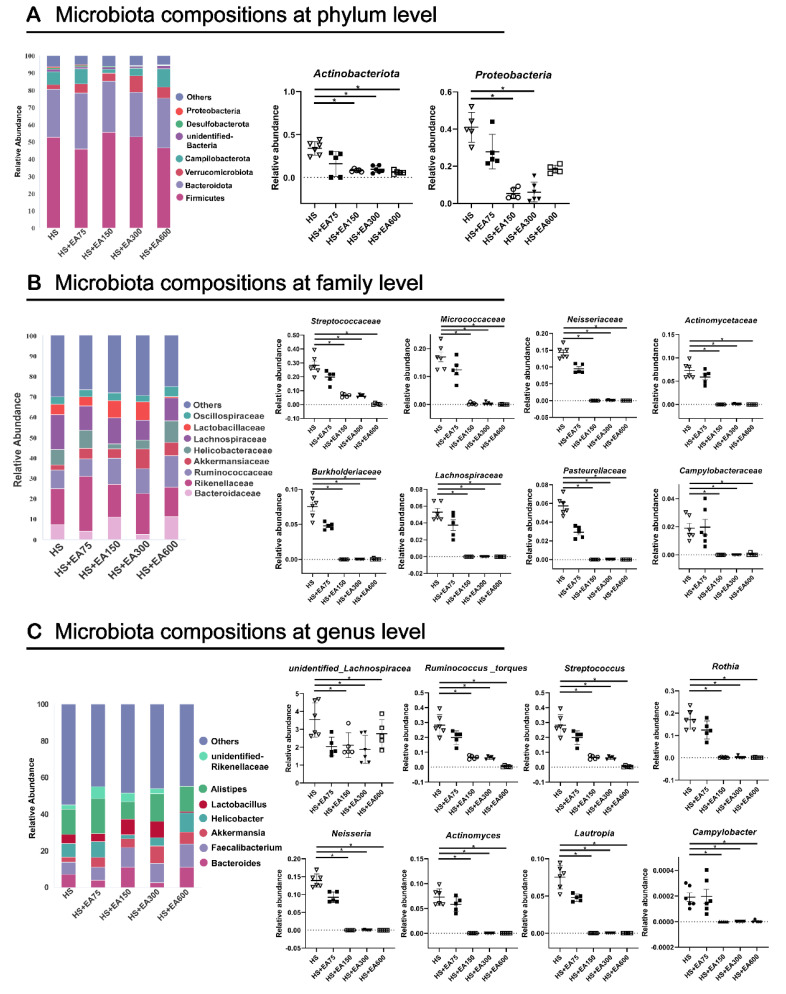
Effects of different levels of EA on cecum microbiota composition at phylum, family, and genus level of heat-stressed broilers. (**A**) Cecum microbiota composition at phylum level; (**B**) Cecum microbiota composition at family level; (**C**) Cecum microbiota composition at genus level; The broilers in the 5 experimental groups were supplemented with basal diets containing different levels of EA (0, 75, 150, 300, and 600 mg/kg) at HS temperature (35 ± 2 °C), which are represented as HS, HS + EA75, HS + EA150, HS + EA300, and HS + EA600, respectively. “*” indicates a significant difference between groups where *p* < 0.05.

**Figure 10 animals-12-01180-f010:**
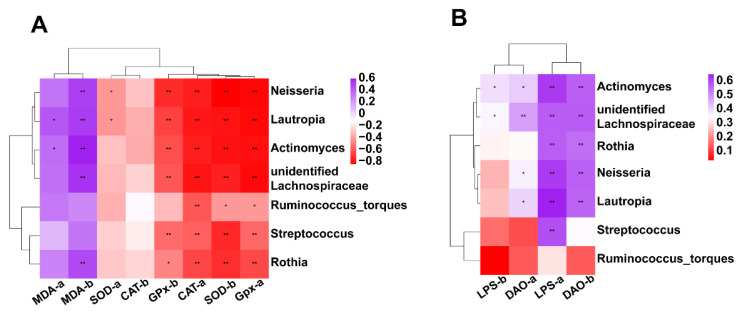
Correlation coefficient and significant analysis between gut microbiota (at genus level) and the parameters of antioxidant capacity or intestinal permeability of the heat-stressed broilers (**A**) Correlation between intestinal microbiota (at genus level) and antioxidant activity. “a” represents the parameters on day 28, and “b” represents the parameters on day 42. (**B**) Correlation between intestinal microbiota (at genus level) and permeability. “a” represents the parameters of 28-day-old broilers, and “b” represents the parameters of 28-day-old broilers. The purple to red color represents an axis from positive correlation to negative correlation. “*” indicates a significant difference between groups where *p* < 0.05. “**” indicates a significant difference between groups where *p* < 0.01.

## Data Availability

The original contributions presented in the study are included in the article/Supplementary Material. Further inquiries can be directed to the corresponding authors.

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
