# Peer review of "Ellagic Acid Improves Antioxidant Capacity and Intestinal Barrier Function of Heat-Stressed Broilers via Regulating Gut Microbiota"

_animals, 2022, doi:10.3390/ani12091180_

Round 1
Reviewer 1 Report
Overall, the paper titled “Ellagic Acid Improves Antioxidant Capacity and Intestinal Barrier Function of Heat-stressed Broilers via Regulating Gut Microbiota” is interesting study, which was well designed. Some issues must be however elucidated before it can be recommended for publication.
Specific remarks
1. line 88. Please be more specific regarding period of birds raising (with or without 14-day adaptation period)
2. line 93 vs. 96. This sentence is confusing; did the birds have unrestricted access to feed or were fed twice per day?
3. line 133-136. Did you record mortality? Why there is no data regarding feed conversion ratio? In fact, this is the most important indicator regarding performance response to treatment. This must be included and discussed in the results and discussion sections.
4. lines 137-146. Give more details regarding analyzes conditions i.e., Spectro-range of detection, wavelength or so…
5. lines 152-158. More details regarding transcript level determination must be given. i.e., how did you choose reference gene (this should be validated for each specific condition (PCR condition or tissue – its expression level and stability). Why only 1 reference gene was considered?
Author Response
Dear reviewer,
Thanks for your great efforts and valuable comments on our manuscript. We have tried our best to improve our manuscript by taking your suggestion. Our responses to your comment are listed below.
Response to Reviewer 1’s Comments
Point 1: line 88. Please be more specific regarding period of birds raising (with or without 14-day adaptation period).
Response 1: Thanks for your comment. During the experiment, after a 14-day adaptation period (feeding with the basal diet), all the broilers were raised for additional 28 days. Accordingly, we have revised the statement in the 2nd round manuscript (Lines 99-100 in the marked vision).
Point 2: line 93 vs. 96. This sentence is confusing; did the birds have unrestricted access to feed or were fed twice per day?
Response 2: Thanks for your comment. Indeed, all broilers had ad libitum access to the experimental diet and water throughout the experiment. In our experiment, to ensure ad libitum feed intake, the broilers were fed enough diet twice per day at 07:30 and 16:30, and had ad libitum water access via pressurized nipple drinkers. I have clarified this point in our revised manuscript (Lines 104-109).
Point 3: line 133-136. Did you record mortality? Why there is no data regarding feed conversion ratio? In fact, this is the most important indicator regarding performance response to treatment. This must be included and discussed in the results and discussion sections.
Response 3: Thanks for your comment. In the experiment, the mortality of birds was recorded when the death occurred (lines 114-115). In our experiment, all broilers were healthy and performed well, no mortality occurred in the NC group. The mortality rate in the HS group was 10% (6 out of 60 chickens). The mortality rate of the birds was below 6% in broilers (14 out of 240 birds) in the dietary EA supplementation groups. We have added this description in the revised manuscript (lines 235-237).
Besides, the feed conversion ratio has been added to Figure3 in the revised manuscript as you suggested and discussed in the results and discussion sections.
Point 4: lines 137-146. Give more details regarding analyzes conditions i.e., Spectro-range of detection, wavelength or so…
Response 4: Thanks for your comment. Thank you very much for your suggestion, I have added details of the analysis conditions in the supplementary materials (Table S2).
Point 4: lines 152-158. More details regarding transcript level determination must be given. i.e., how did you choose reference gene (this should be validated for each specific condition (PCR condition or tissue – its expression level and stability). Why only 1 reference gene was considered?
Response 5: Thanks for your comment. According to previously published studies (Wan. et, al. 2018; Nawab. et, al. 2019), under heat stress conditions, β-actin is usually used for the normalization of gene expression. In addition, our experimental results also showed that the cycle threshold (Ct) values were relatively consistent between the control and heat-stressed group. Together, it can be inferred that the mRNA level of β-actin is not affected by heat stress conditions. Thus, we chose the β-actin as a housekeeping gene in our study.
References:
Wan, X., Ahmad, H., Zhang, L., Wang, Z., & Wang, T. (2018). Dietary enzymatically treated Artemisia annua L. improves meat quality, antioxidant capacity and energy status of breast muscle in heat-stressed broilers. Journal of the science of food and agriculture, 98(10), 3715–3721. https://doi.org/10.1002/jsfa.8879
Nawab, A., Li, G., An, L., Wu, J., Chao, L., Xiao, M., Zhao, Y., Birmani, M. W., & Ghani, M. W. (2019). Effect of curcumin supplementation on TLR4 mediated non-specific immune responses in liver of laying hens under high-temperature conditions. Journal of thermal biology, 84, 384–397. https://doi.org/10.1016/j.jtherbio.2019.07.003
Thanks again for your comments. If you have any questions, please do not hesitate to contact us at the address below.
Yours sincerely,
Qian Jiang, Professor, College of Animal Science and Technology, Hunan Agricultural University
Email: jiangqian@hunau.edu.cn
Reviewer 2 Report
The Manuscript entitled “Ellagic Acid Improves Antioxidant Capacity and Intestinal Bar- 2 rier Function of Heat-stressed Broilers via Regulating Gut Mi- 3 crobiota”, by Tai Yang et al., investigated the effects of dietary ellagic acid on the antioxidant system, gut barrier function, and gut microbiota of heat stressed broilers. The Manuscript is original, well designed and written. I support its publication after the modifications listed below; in particular, major revisions are required in relation to the Analysis of intestinal Microbial community (16S metabarcoding).
Line 24: replace “of different doses” with “at different doses”
Introduction: The introduction is quite short, I suggest to improve and expand it with reference to the studies using EA in other food animals and in broilers for different purposes (e.g. effects of EA supplementation on semen quality, effects of EA in enhancing productive performance and alleviating oxidative stress in quail etc.)
Line 87: replace “of different doses” with “at different doses”
Lines 103-104: replace “was shown” with “is shown” and “were described” with “are described”
Line 161: Just a suggestion for future experiments, there is actually a better kit for the isolation of microbial DNA from stool and gut samples, that is the QIAamp Power Fecal Pro DNA Kit; it should improve the amount of microbial DNA extracted.
Line 164: Please specify which regions of the 16S rRNA gene were amplified, the primers used, the kit used to purify PCR amplicons and the index kit used for library preparation.
Lines 178-180: This paragraph should be removed from the results section, as it is part of the study design.
Figures 2-3-4-5-6: I would suggest to use different colors instead of different patterns for bars, as they are confusing.
Line 280: As a general comment, the 16S metabarcoding results reach reliable classification level at family level; the genus level might contain mistakes, and the resolution is strictly related to the regions amplified, which here the authors did not specify. Thus, results should be carefully interpreted, and the M&M section related to the 16S metabarcoding should be improved.
Line 280: Data obtained preforming the 16S rRNA sequencing (metabarcoding) to study the microbial population should be submitted in one or more online repositories (e.g. NCBI). Please report the chosen repository/repositories and the accession number(s).
Line 323: there is a typo, “using PICRUSt2”
Line 373: replace “Similar to a recent finding that” with “Similar to a recent finding showing that”
Line 446: Lactobacillus is not an order; please remove the word “order”.
Author Response
Dear reviewer,
Thanks for your great efforts and valuable comments on our manuscript. We have tried our best to improve our manuscript by taking your suggestion. Our responses to your comment are listed below.
Response to Reviewer 2’s Comments
Point 1: Line 24: replace “of different doses” with “at different doses”
Response 1: Thanks for your suggestion. I have made changes according to your suggestions.
Point 2: Introduction: The introduction is quite short, I suggest to improve and expand it with reference to the studies using EA in other food animals and in broilers for different purposes (e.g. effects of EA supplementation on semen quality, effects of EA in enhancing productive performance and alleviating oxidative stress in quail etc.)
Response 2: According to your suggestions, we have strengthened the introduction section. relevant literature and added it to the article (Lines 46-49, 71-78).
Point 3: Line 87: replace “of different doses” with “at different doses”.
Response 3: According to your suggestions, we have revised them.
Point 4: Lines 103-104: replace “was shown” with “is shown” and “were described” with “are described”
Response 4: According to your suggestions, we have revised them.
Point 5: Lines 103-104: replace “was shown” with “is shown” and “were described” with “are described”
Response 5: According to your suggestions, we have made a revision.
Point 6: Line 161: Just a suggestion for future experiments, there is actually a better kit for the isolation of microbial DNA from stool and gut samples, that is the QIAamp Power Fecal Pro DNA Kit; it should improve the amount of microbial DNA extracted.
Response 6: Thank you very much for your valuable suggestions. We will use the QIAamp Power Fecal Pro DNA Kit in our further experiment.
Point 7: Line 164: Please specify which regions of the 16S rRNA gene were amplified, the primers used, the kit used to purify PCR amplicons and the index kit used for library preparation.
Response 7: Thanks for your comments. In the revised manuscript, we have specified the information in terms of region, primers, and the kit for the library preparation (Lines 183-199).
Point 8: This paragraph should be removed from the results section, as it is part of the study design
Response 8: Thanks for your suggestion. We have removed it from the results section, please check it.
Point 9: Figures 2-3-4-5-6: I would suggest to use different colors instead of different patterns for bars, as they are confusing.
Response 9: As you suggested, in the revised manuscript, we used different colors for the figures’ presentation.
Point 10: As a general comment, the 16S metabarcoding results reach reliable classification level at family level; the genus level might contain mistakes, and the resolution is strictly related to the regions amplified, which here the authors did not specify. Thus, results should be carefully interpreted, and the M&M section related to the 16S metabarcoding should be improved.
Response 10: Thanks for your comments. In the revised manuscript, we have specified the 16S regions and added the data analysis at the family level (Figure 9).
Point 11: Data obtained preforming the 16S rRNA sequencing (metabarcoding) to study the microbial population should be submitted in one or more online repositories (e.g. NCBI). Please report the chosen repository/repositories and the accession number(s).
Response 11: Data obtained performing the 16S rRNA sequencing (metabarcoding) to study the microbial population has been submitted to NCBI -Ellagic Acid regulates gut microbial on Heat-stressed Broiler. And this Sequence Read Archive (SRA) submission will be released on publication, the accession number is SUB11337682.
Point 12: there is a typo, “using PICRUSt2”
Response 12: Thanks for your reminder, we have revised this typo.
Point 13: replace “Similar to a recent finding that” with “Similar to a recent finding showing that”
Response 13: Thanks for your comments. I replace “Similar to a recent finding that” with “Similar to a recent finding showing that”
Point 14: Lactobacillus is not an order; please remove the word “order”
Response 14: Thank you very much for your suggestion. I have removed the word “order”
Thanks again for your comments. If you have any questions, please do not hesitate to contact us at the address below.
Yours sincerely,
Qian Jiang, Professor, College of Animal Science and Technology, Hunan Agricultural University
Email: jiangqian@hunau.edu.cn
Round 2
Reviewer 1 Report
This paper has been significantly improved. I have no more remarks.
Reviewer 2 Report
The article "Ellagic Acid Improves Antioxidant Capacity and Intestinal Barrier Function of Heat-stressed Broilers via Regulating Gut Microbiota" is now ready to be published in the present, revised form.